# Silver Nanoparticle–Antibiotic Combinations: A Strategy to Overcome Bacterial Resistance in *Escherichia coli*, *Salmonella* Enteritidis and *Staphylococcus aureus*

**DOI:** 10.3390/antibiotics14100960

**Published:** 2025-09-24

**Authors:** Mariana Homem de Mello Santos, Thiago Hideo Endo, Sara Scandorieiro, Wander Rogério Pavanelli, Renata Katsuko Takayama Kobayashi, Gerson Nakazato

**Affiliations:** 1Laboratory of Basic and Applied Bacteriology, Department of Microbiology, Center of Biological Sciences, Universidade Estadual de Londrina, Londrina 86057-970, Brazil; mellomariana28@uel.br (M.H.d.M.S.); th.endo@uel.br (T.H.E.); kobayashirkt@uel.br (R.K.T.K.); 2Laboratory of Innovation and Cosmeceutical Technology, Department of Pharmaceutical Sciences, Center of Health Sciences, Hospital Universitário de Londrina, Londrina 86038-350, Brazil; sarascandorieiromicro@gmail.com; 3Laboratory of Experimental Protozoology, Department of Pathological Sciences, Center of Biological Sciences, Universidade Estadual de Londrina, Londrina 86057-970, Brazil; wanderpavanelli@uel.br

**Keywords:** nanotechnology, synergism, combination therapy and mechanism of action

## Abstract

**Background/Objectives**: Bacterial resistance to antimicrobials is a major global health challenge, limiting the effectiveness of conventional therapies and complicating infection control. The aim of this study was to investigate the antibacterial potential of biologically synthesized silver nanoparticles (Bio-AgNP), alone and in combination with ampicillin (AMP) and enrofloxacin (ENRO), against multidrug-resistant (MDR) bacterial isolates of clinical and veterinary relevance. **Methods:** The antibacterial activity of Bio-AgNP, AMP, and ENRO, alone or in combination, was assessed against reference strains and MDR isolates of *Escherichia coli*, *Salmonella enterica* serovar Typhimurium and Enteritidis, and *Staphylococcus aureus*. Minimum inhibitory concentration (MIC) values were determined, and bacterial tolerance to prolonged antimicrobial exposure was evaluated. Additionally, assays were conducted to explore potential mechanisms of action, including cell membrane permeability and oxidative stress induction. **Results**: All bacterial strains developed increased MIC values after prolonged exposure to conventional antibiotics, confirming resistance. Only *E. coli* developed resistance to Bio-AgNP. Notably, the Bio-AgNP + AMP combination effectively restored susceptibility in *E. coli*, while only *S. Enteritidis* developed resistance to this combination upon prolonged exposure. The synergistic effect of Bio-AgNP with conventional antibiotics significantly reduced bacterial growth within two hours, compared with longer times observed in monotherapy. Mechanistic analysis suggested that the combinations increased membrane permeability, facilitating antibiotic entry. **Conclusions**: Bio-AgNPs combined with AMP or ENRO enhanced antibacterial activity and overcame resistance in MDR isolates, representing a promising therapeutic alternative. The biological synthesis of Bio-AgNPs, capped with organic biomolecules, supports their potential as safe adjuvants to conventional antibiotics in combating MDR bacterial infections.

## 1. Introduction

Bacterial resistance has emerged as a major global public health issue, as declared by the World Health Organization (WHO). Bacterial infections, once easily treatable, are now becoming increasingly difficult to cure, leading to more severe diseases, higher incidence rates, and increased treatment costs [1]. The excessive and widespread use of antimicrobials in healthcare, veterinary, and agricultural sectors contribute to bacterial resistance by exerting selective pressure on microorganisms, in line with Darwinian theory of natural selection [2]. It is estimated that bacterial resistance causes 1.27 million deaths in 2019 and was associated with 4.95 million additional deaths [3]. Consequently, the WHO encourages researchers to develop novel antimicrobials and implement action plans to promote the discovery of effective and safe antimicrobial agents with multiple mechanisms of action [1]. The most studied nanoparticles as antimicrobials are silver nanoparticles (AgNPs). Their unique electrical, optical, and catalytic properties, especially their antibacterial action against multidrug-resistant (MDR) bacteria, have drawn significant attention, largely due to their multiple mechanisms of action and large surface area-to-volume ratio [4,5]. As a result, AgNPs emerge as an alternative to conventional antimicrobials and can be applied in diverse ways, such as in the decontamination of medical supplies, food packaging, and the treatment of infections [6]. AgNPs can be mainly synthesized through chemical, biological and physical methods. The chosen synthesis route impacts on their properties, such as size, stability, ion release capacity, biological effects, and toxicity [5]. For instance, biological synthesis utilizes bacteria, fungi, or plant extracts as reducing agents, making it a promising, eco-friendly, and low-cost alternative [7]. Thus, careful consideration must be given to the synthesis method to maximize AgNP antibacterial efficacy while reducing cytotoxicity [8]. The antibacterial action of AgNPs is not yet fully understood, but studies suggest it can be attributed to three mechanisms: (1) adhesion to the bacterial cell surface, altering cell wall and membrane properties; (2) entry into the cell and interaction with proteins and DNA; and (3) the release of silver ions, which interact with cellular components and disrupt metabolic pathways [9]. These multiple mechanisms have generated optimism that AgNPs may be a challenging antimicrobial for bacteria to develop resistance. However, studies have shown that bacteria can develop resistance to AgNPs through genomic alterations [10,11]. Conversely, research indicates that combining AgNPs with various compounds, such as conventional antibiotics, plant extracts, or polymers, can produce additive or synergistic effects. The discovery of antibiotics represented an incredible medical breakthrough in the 20th century, referred to as the ‘Golden Age of Antibiotics’. Their introduction into clinical practice significantly reduced the morbidity and mortality related to bacterial infections. As a result, many antibiotics lost their effectiveness due to bacterial resistance mechanisms, creating an urgent need for the development of alternative antimicrobials. Bacterial resistance has affected various classes of antibiotics, including beta-lactams and fluoroquinolones, which interfere with bacterial cell wall synthesis and DNA replication, respectively [12]. Resistance mechanisms to ampicillin, widely used in human medicine to treat urinary and respiratory infections, salmonellosis, sepsis, and meningitis, have already been reported [13,14]. Enrofloxacin, a fluoroquinolone exclusively used in veterinary medicine to treat a variety of bacterial infections, such as skin, urinary, digestive, and respiratory infections, has also been reported concerning its bacterial resistance mechanisms [15,16]. One strategy to combat bacterial resistance to conventional antibiotics relies on their combination with AgNPs. Studies have shown that such combinations are not associated with the emergence of resistance, possibly due to the different mechanisms of action of the two compounds [17,18]. Combination therapies are preferred over monotherapy, as they broaden the spectrum of empirical treatments, enhance efficacy through synergy, reduce dosage to minimize host toxicity, and lower the likelihood of selecting resistant strains [19]. AgNPs can exhibit cytotoxic effects on human cells depending on their concentration. A key insight for their clinical use is that combining them with conventional antibiotics allows for effective treatment at reduced doses [20]. Therefore, this study aimed to evaluate the antibacterial activity and elucidate the potential mechanisms of action of the binary combination of biogenic AgNPs with ampicillin and enrofloxacin. AgNPs can enhance the entry of enrofloxacin, allowing it to reach its target more effectively, while ampicillin facilitates the uptake of AgNPs into the cell. This synergy amplifies the antimicrobial effects. The objective is to restore the effectiveness of these antibiotics, which has reduced due to the increasing prevalence of resistant bacterial strains.

## 2. Results

### 2.1. Minimal Inhibitory Concentration and Minimal Bactericidal Concentration

The MIC and MBC values obtained for the reference strains and MDR are shown in Table 1.

### 2.2. Checkerboard Assay

The checkerboard assay revealed a reduction in MIC values for both tested combinations as shown in Figure 1. The details of the MIC values in combination, FICI, and percentage reduction are provided in the Appendix A.

### 2.3. Time–Kill

*E. coli* reference strain treated with C1 and C2 displayed no growth after 2 h, whereas treatment with ampicillin and enrofloxacin alone resulted in growth inhibition after 4 and 7 h, respectively (Figure 2). These results highlight the improved efficacy of conventional antibiotics when combined with Bio-AgNP.

### 2.4. Prolonged Exposure of Bacteria to Antimicrobials

The bacterial strains were continuously exposed to individuals and combined antimicrobials to assess their tolerance. After prolonged exposure, the MIC values were determined to evaluate any increase (Appendix A) and to analyze the sensitivity profile of the strains to ampicillin and enrofloxacin, confirming whether resistance had developed. As expected, the standard strains developed resistance to ampicillin and enrofloxacin after a week of exposure. Regarding Bio-AgNPs, only *E. coli* developed resistance, with the MIC increasing from 31.5 µM to 2000 µM, indicating a notable rise in bacterial tolerance.

In addition to enhancing antimicrobial efficacy, the combinations were also capable of preventing resistance, except for C1 against *S*. Enteritidis. Notably, although *E. coli* developed resistance to individual Bio-AgNP, C1 was still able to enhance antimicrobial activity and overcome resistance. Table 2 shows the increase in MIC values for individual antimicrobials after prolonged exposure. Table 3 and Table 4 present the MIC values for combinations after prolonged exposure.

### 2.5. Analysis of Cross-Resistance

The antibiogram performed before prolonged exposure to the treatments showed that all three reference strains were susceptible to all tested antibiotics from different classes (tables are in the Appendix A).

The susceptibility profile of bacterial strains after prolonged exposure to the treatments (Table 5, Table 6 and Table 7) revealed cross-resistance to antibiotics from different classes, suggesting a possible interaction between resistance mechanisms acquired through prolonged exposure. Although some samples exhibited a reduction in the inhibitory zone diameter, indicating decreased sensitivity, this was not sufficient to characterize resistance according to the zone diameter breakpoints established by CLSI.

An intriguing result was that *S.* Enteritidis treated with C2, and *S. aureus* treated with C1 and C2 did not exhibit cross-resistance to any antibiotic, indicating superior efficacy in preventing the selection of resistant strains. This result may be associated with the synergy between antimicrobials and the potent antimicrobial activity of Bio-AgNP shown in this study.

### 2.6. FTIR Analysis of Bio-AgNP Combined with Conventional Antibiotics

FTIR analysis was conducted on conventional antibiotics alone, Bio-AgNP, and their combinations. The FTIR transmittance profiles of the Bio-AgNP–antibiotic treatments displayed characteristic peaks of both Bio-AgNP and the conventional antibiotics (Figure 3). Nevertheless, the spectra did not indicate any chemical bonds between the compounds, as the peaks of the combinations were identical to those of the individual treatments. Thus, this suggests that the mechanism of action of the compounds is independent, and the combined interaction of the antimicrobials results in either additive or synergistic effects. Furthermore, Bio-AgNP alters membrane permeability, facilitating the entry of standard antibiotics and enhancing their action on specific targets.

### 2.7. Alterations in Cytoplasmic Membrane Permeability

In the dye uptake assay, rose Bengal revealed that bacteria treated with C1 showed statistically higher dye absorption rates—27.86% (*p*-value < 0.01), compared to the untreated control group (CTL), which presented absorption rate of 12.28%. However, in the Evans blue dye assay no statistical difference was found between treated bacteria and the CTL. Although some treatments did not show statistically significant differences, they differed in terms of absolute values, which is biologically relevant. The bacterial cells exposed to both Evans blue and rose Bengal showed numerically higher absorption values when treated with bio-AgNP, C1, and C2. These results suggest that Bio-AgNP, C1, and C2 could slightly alter membrane permeability compared to the CTL group, as indicated by the increased dye uptake in these treatments. The results of the experiment were validated through the statistical difference between the negative and positive (Triton X-100) controls (Figure 4).

In the analysis of biomolecule leakage (Figure 5), bacterial cells treated with Bio-AgNP (30 ng/µL) and C2 (35 ng/µL) compared to the control (CTL) (15 ng/µL), exhibited a statistically significant release of dsDNA, whereas no treatment caused a significant leakage of total proteins. Nevertheless, consistent with the results observed in the dye uptake assay, treatments with Bio-AgNP, C1, and C2 showed a trend toward increased leakage for both biomolecules, with numerically higher values compared to their respective control groups. Triton X-100, used as a positive control, caused the leakage of 2000 ng/µL of dsDNA and 30 mg/mL of total proteins, confirming alterations in membrane permeability.

### 2.8. Detection of ROS

Figure 6 shows the detection of ROS in treated and untreated *E. coli* ATCC 25922. Fluorescence was measured in arbitrary units (AU) using excitation at 495 nm and emission at 520 nm. Hydrogen peroxide (H_2_O_2_) was used as a positive control and demonstrated an increase in ROS detection over the 2-h period when compared to the negative control, thus validating the experiment. The negative control consisted of PBS alone, which detected basal ROS levels in the cells, starting at 64.25 AU and ending at 93.90 AU after 2 h.

When comparing the individual treatments, it was observed that ampicillin and enrofloxacin exhibited values close to the negative control throughout the readings, confirming that these treatments do not operate via oxidative stress. After 2 h, ampicillin showed 93.12 AU, enrofloxacin showed 91.60 AU, and the negative control was 93.90 AU, demonstrating that the absolute values were very similar. However, no detectable increase in ROS level was observed in treatments with ampicillin and enrofloxacin. Bio-AgNP exhibited lower values compared to the negative control in all readings. After 2 h, Bio-AgNP measured 80.22 AU, while the negative control was 93.90 AU, indicating that this compound reduces ROS production. All time points showed a statistically significant difference when comparing the negative control (PBS) with Bio-AgNP.

Combinations 1 and 2 yielded results like those of Bio-AgNP, with lower values when compared to the negative control. After 2 h, combination 1 measured 82.45 AU, combination 2 measured 81.37 AU, and the negative control was 93.90 AU. This result showed that Bio-AgNP, even when combined with conventional antibiotics, could reduce ROS production.

### 2.9. Inhibition of Efflux Pump

The MIC of tetracycline (TET) for *E. coli* XL1-blue was 312.5 μg/mL. Upon combination with C1, C2, and Bio-AgNP at subinhibitory concentrations, the MIC of TET remained the same. Similarly, the MIC of TET for *E. coli* K12 was 0.25 μg/mL, and it remained unchanged (considering the standard deviation). Thus, these compounds do not inhibit the efflux pump since there was no decrease in the MIC values for TET in the presence of treatments.

## 3. Discussion

The present study provides a comprehensive evaluation of the antimicrobial potential of biologically synthesized silver nanoparticles (Bio-AgNPs), both alone and in combination with conventional antibiotics, against reference and multidrug-resistant (MDR) bacterial strains. The results demonstrate that Bio-AgNPs possess broad-spectrum activity and that their combinations with ampicillin (C1) or enrofloxacin (C2) can enhance antimicrobial efficacy and reduce resistance development.

Prolonged exposure of standard strains to ampicillin and enrofloxacin led to the development of resistance, highlighting the well-documented risk associated with sublethal concentrations of antibiotics, which may induce genetic mutations and promote the emergence of resistant phenotypes [21]. Clinically, such subinhibitory exposures can occur due to incomplete treatments, missed doses, or limited drug penetration in tissues like bone or cerebrospinal fluid [22]. Interestingly, although *E. coli* developed resistance to Bio-AgNP alone, combination therapies—especially C1—were able to maintain or enhance antimicrobial activity and prevent the selection of resistant strains in most cases, suggesting a protective effect of the combined treatment against resistance emergence [22,23,24]. These findings are consistent with previous reports indicating that nanoparticle–antibiotic combinations can exhibit synergistic or additive effects, mitigating resistance selection and improving therapeutic outcomes [25].

The antibiogram performed before and after prolonged exposure also revealed that combination treatments were more effective at preventing cross-resistance compared to individual treatments. Notably, *S.* Enteritidis treated with C2 and *S. aureus* treated with C1 or C2 did not exhibit cross-resistance to any tested antibiotic. Cross-resistance may arise from mutations affecting multiple genes, including those associated with efflux systems or antibiotic targets, and can be promoted by subinhibitory antimicrobial concentrations [21]. These findings indicate that rational selection of combination therapies can limit the emergence of cross-resistance, supporting previous studies that demonstrate similar benefits of synergistic and additive combinations [22,26].

FTIR analysis confirmed that Bio-AgNPs do not form chemical bonds with conventional antibiotics, as the characteristic peaks of the combinations were identical to those of the individual compounds. This suggests that the mechanisms of action of Bio-AgNPs and antibiotics remain independent, and the observed additive or synergistic effects arise from complementary biological interactions rather than chemical modifications of the molecules. One plausible mechanism is the alteration of bacterial membrane permeability. Dye uptake assays and the analysis of intracellular biomolecule leakage revealed that Bio-AgNP, C1, and C2 slightly increased bacterial membrane permeability, as evidenced by higher rose Bengal uptake and numerically elevated release of dsDNA and proteins. These results align with prior studies demonstrating that AgNPs accumulate on bacterial membranes, induce structural abnormalities, create membrane depressions, and facilitate intracellular access of antimicrobials [17,23,24,25,27,28,29,30,31,32]. The ability of Bio-AgNP to enhance the intracellular entry of antibiotics likely underlies the synergistic and additive effects observed, particularly against MDR strains.

Treatments with Bio-AgNP, either alone or in combination, consistently resulted in lower ROS levels compared to the negative control; thus, we hypothesize an antioxidant rather than oxidative effect. This phenomenon is likely related to the synthesis of nanoparticles from *Trichilia catigua* (catuaba) extract, which contains bioactive phytochemicals such as flavonoids, tannins, alkaloids, and saponins [33,34]. These compounds confer potent antioxidant activity and their presence on the nanoparticle surface may neutralize free radicals, as reported for other plant-derived nanoparticles [35,36]. The reduction of ROS, while seemingly contrary to the classical mechanism of AgNP toxicity, does not impair the antimicrobial efficacy of Bio-AgNPs and may confer additional benefits by reducing potential oxidative damage to host tissues.

The efflux pump inhibition assay demonstrated that neither Bio-AgNPs nor their combinations interfere with efflux pump activity in *E. coli*. Efflux pumps are major bacterial resistance determinants, and their inhibition can enhance susceptibility to antimicrobials. Previous studies reported that AgNP can partially restore the activity of antibiotics in *S. aureus* by inhibiting efflux pumps, particularly in isolates from bovine mastitis [37]. However, the absence of efflux pump inhibition in the present study indicates that the observed antimicrobial effects are mediated primarily by other mechanisms, such as membrane permeability alteration, rather than classical efflux pump interference.

Overall, the data presented here underscore the therapeutic potential of Bio-AgNP, particularly when used in combination with conventional antibiotics. The combined treatments demonstrated the ability to enhance antimicrobial efficacy, prevent resistance development, and act through mechanisms including membrane permeability alteration rather than oxidative stress or efflux pump inhibition. These findings contribute to a growing body of evidence supporting the design of novel antimicrobial strategies that integrate biologically synthesized nanoparticles with conventional drugs to combat multidrug-resistant pathogens.

## 4. Materials and Methods

### 4.1. Bacterial Strains

The bacterial reference strains used in the experiments were obtained from the American Type Culture Collection (ATCC). The reference strains and MDR isolates were maintained in Brain Heart Infusion (BHI) medium with 25% glycerol at −80 °C in the Laboratory of Basic and Applied Bacteriology (LBBA), Department of Microbiology, Center for Biological Sciences, State University of Londrina.

The reference strains tested were *Escherichia coli* ATCC 25922, *Escherichia coli* XL1-Blue, *Escherichia coli* K12, *Salmonella enterica* serovar Enteritidis ATCC 13076, and *Staphylococcus aureus* ATCC 25923. Additionally, multidrug-resistant (MDR) strains were included in the study: a veterinary MDR isolate of *Salmonella enterica* serovar Typhimurium, and clinical MDR isolates of *Escherichia coli* and *Staphylococcus aureus*.

### 4.2. Antimicrobials

The conventional antibiotics used were ampicillin and enrofloxacin, both obtained from Sigma-Aldrich (St. Louis, MO, USA). The ampicillin stock solution was prepared at 20 mg/mL (80% purity); the powder was dissolved in ultrapure water, filtered through a sterilized 0.22 μm membrane, and stored at −4 °C. Enrofloxacin was prepared at 0.01 mg/mL by dissolving 1 mg of the powder in 1 mL of dimethyl sulfoxide (DMSO) at 50 °C, followed by a 1:10 dilution in deionized water (10% DMSO in final solution).

The alternative antimicrobial agent used was the biogenic silver nanoparticle (Bio-AgNP) Nanoverde, produced by GRAL Bioativos^®^ LTDA, Londrina, Brazil. This Bio-AgNP was synthesized after the reduction of AgNO_3_ by *Trichilia catigua* Adr. Juss bark aqueous extract. Bio-AgNP has a size of 82.73 nm, a zeta potential of −23.23 mV, and a polydispersity index (PDI) of 0.17. The biosynthesis and characterization were carried out according to the patent methodology BR1020210163755 [38].

### 4.3. Antibacterial Activity

#### 4.3.1. MIC and MBC Determination

The Minimum Inhibitory Concentration (MIC) was determined following the guidelines outlined in document M07 by the Clinical and Laboratory Standards Institute [39]. Antimicrobial concentrations were prepared in Mueller-Hinton (MH) broth, and the MIC was defined as the minimum concentration that inhibited visible growth after 24 h of treatment at 37 °C.

The Minimum Bactericidal Concentration (MBC) was determined by subculturing 10 μL from the MIC and higher concentrations onto MH agar plates. The MBC was defined as the lowest concentration that killed ≥99.9% of bacterial cells after 24 h of antimicrobial treatment, following the guidelines provided by the National Committee for Clinical Laboratory Standards (NCCLS, 1999) [40].

#### 4.3.2. Checkerboard Assay

The antibacterial interaction of the active compounds in combination was assessed using broth microdilution, following a modified version of the methodology described by Domínguez (2023) [41]. Various concentrations of both antimicrobials were tested. The combinations evaluated were ampicillin + Bio-AgNP (C1) and enrofloxacin + Bio-AgNP (C2). The assay was performed in triplicate using a 96-well plate. For C1, Bio-AgNP was serially diluted along the X-axis (ranging from 500 μM to 1.95 μM), while ampicillin was diluted along the Y-axis (ranging from 64 μg/mL to 0.25 μg/mL), creating a double concentration gradient of both agents. For C2, the same process was performed—Bio-AgNP concentration was kept constant, and enrofloxacin was diluted from 0.06 μg/mL to 0.0005 μg/mL.

Bacterial cultures were standardized in sterile saline solution (0.85%) to the 0.5 McFarland turbidity standard, corresponding to 1.5 × 10^8^ CFU/mL, and subsequently diluted (1:100) in MH broth before inoculation into the wells, achieving a final concentration of 7.5 × 10^5^ CFU/mL. Sterility and growth controls were included. The 96-well plates were incubated at 37 °C for 24 h.

The fractional inhibitory concentration (FIC) of each antimicrobial was calculated by dividing the MIC observed in combination by the MIC of the individual compound, as shown in the following equation:FIC = (MIC combination)/(MIC individual)

The FIC index (FICI) was determined by adding the two FIC values. A FICI value ≤ 0.5 indicates a synergistic interaction; 0.5 to 1.0 indicates an additive interaction. Interactions with FICI values ranging from 1.0 to 2.0 are considered indifferent, while values > 2.0 indicate antagonism. The FICI was calculated as follows:FICI = FIC (1st antimicrobial) + FIC (2nd antimicrobial)

#### 4.3.3. Time–Kill

The time–kill assay was performed to assess bacterial kinetics in response to both combined and individual antimicrobial agents, following the modified guidelines outlined in document M26-A by the NCCLS [40]. Treatments included combination 1 (ampicillin + Bio-AgNP), combination 2 (enrofloxacin + Bio-AgNP), ampicillin, enrofloxacin, Bio-AgNP at MIC, and a growth control containing only Mueller-Hinton broth. Bacterial inoculum was added to each treatment to reach a final concentration of 5 × 10^5^ CFU/mL, and samples were incubated at 37 °C.

At designated time points (0, 0.5, 2, 4, 7, 10, and 24 h), 100-μL aliquots were taken from each treatment, and serial dilutions were performed in 900 μL of sterile saline (0.85%). From these dilutions, 10-μL aliquots were plated onto MH agar in triplicate. The plates were incubated at 37 °C for 24 h, after which colony counts were performed to determine the bacterial load (CFU/mL) at each time point.

#### 4.3.4. Prolonged Exposure of Bacteria to Antimicrobials

Extended antimicrobial exposure was carried out over 30 consecutive days. The bacterial strains *E. coli* ATCC 25922, *S.* Enteritidis ATCC 13076, and *S. aureus* ATCC 25923 were subjected to treatments involving combinations of antimicrobials and individual antimicrobials; MH broth without antimicrobials served as a control to assess the potential development of tolerance. Treatments were prepared at subinhibitory concentrations (sub-MIC), with the concentration gradually increasing with each daily passage. Specifically, 0.05 mL of the previous day’s culture was transferred to 0.95 mL of MH broth containing the antimicrobials or to the untreated control. The control without antimicrobials ensured that any observed alterations in the treated samples were due to the antimicrobials being tested rather than the maintenance of bacteria in culture. The treatments were incubated in a shaking incubator at 120 rpm and 37 °C for 24 h.

#### 4.3.5. Disk Diffusion Test

The antibiogram was performed before and after extended exposure to determine whether the reference strains developed resistance to ampicillin and enrofloxacin after 30 days of treatment with individual and combined antimicrobials and was used to analyze the development of cross-resistance to other classes of antibiotics.

The antibiotics tested against *S.* Enteritidis and *E. coli* were amoxicillin-clavulanate, ampicillin, cefazolin, cefepime, cefoxitin, ceftriaxone, ciprofloxacin, chloramphenicol, enrofloxacin, fosfomycin, gentamicin, imipenem, nitrofurantoin, sulfamethoxazole-trimethoprim, tetracycline, and tobramycin.

The antibiotics tested against *S. aureus* were ampicillin, azithromycin, cefoxitin, ciprofloxacin, clindamycin, chloramphenicol, enrofloxacin, gentamicin, nitrofurantoin, sulfamethoxazole-trimethoprim, tetracycline, and vancomycin.

Following the CLSI guidelines [42], colonies of recently isolated bacteria, grown for 18 to 24 h, were used. These colonies were suspended in sterile saline solution (0.85% NaCl) until the turbidity corresponded to a 0.5 McFarland standard (1.5 × 10^8^ CFU/mL). A sterile swab was then dipped into the bacterial suspension, and excess fluid was removed by pressing the swab against the inner wall of the tube. The swab was gently streaked across MH agar plates in five different directions to ensure even distribution. After allowing the agar surface to dry for 10 to 15 min, commercial antimicrobial disks were applied using flamed and cooled forceps, with gentle pressure to ensure proper adhesion. The plates were then incubated at 37 °C for 18 to 24 h.

### 4.4. Characterization of Mechanism of Action of Antimicrobials

#### 4.4.1. Dye Absorption Assay

Alterations in membrane permeability were assessed [43,44]. An overnight culture of *E. coli* ATCC 25922 was grown in MH broth under agitation at 120 rpm and 37 °C. The culture was distributed into 1 mL microtubes, centrifuged (Universal 320 R, Hettich, Tuttlingen, Germany) at 8000 rpm for 10 min, and the supernatant was discarded. The bacterial pellet was washed and resuspended in PBS (phosphate-buffered saline, pH 7.4) to remove residual culture medium, followed by another centrifugation under the same conditions. Treatments with C1, C2, ampicillin, enrofloxacin, and Bio-AgNP were prepared at previously determined subinhibitory concentrations. Triton X-100 served as the positive control, and PBS alone as the negative control. The treatments were added to the bacterial suspension and incubated for 3 h at 120 rpm.

After incubation, the samples were centrifuged at 8000 rpm for 10 min, the supernatant discarded, and the bacterial pellet resuspended in PBS containing 0.001% dye. Evans blue and rose Bengal dyes were added to the suspension, followed by a 10-min incubation at 37 °C. A final centrifugation at 8000 rpm for 10 min was performed, and the supernatants were collected. These supernatants were transferred to a 96-well plate and read at 605 nm (Evans blue) and 530 nm (rose Bengal) using a spectrophotometer. The untreated bacterial suspension was used as a negative control, and the dye solutions served as controls to represent the total amount of dye without any bacterial absorption. To determine the percentage of dye absorbed by the bacterial cells, the remaining dye in the supernatant was first calculated as follows:Residual dye (%) = ((Sample absorbance)/(Dye absorbance)) × 100

This value represents the proportion of dye that was not taken up by the bacteria. The percentage of dye absorption was then determined by subtracting this value from 100:Dye absorption (%) = 100 − Residual dye (%)

The test was performed in quadruplicates on four different occasions.

#### 4.4.2. Leakage of Cytoplasmic Contents

Quantification of total DNA and proteins was conducted to analyze the intracellular content in the extracellular medium using the Nanodrop system (Thermo Scientific, Waltham, MA, USA). The *E. coli* inoculum was cultured overnight in MH broth under agitation at 120 rpm and 37 °C. Subsequently, 1 mL aliquots were collected and centrifuged at 8000 rpm for 10 min, and the supernatant was discarded. The bacterial pellet was washed with 1 mL of PBS and centrifuged again under the same conditions. The pellets were then resuspended in treatments at previously determined subinhibitory concentrations and incubated for 3 h under agitation at 120 rpm. Triton X-100 was used as the positive control for membrane permeability alteration, while PBS alone served as the negative control. After incubation, a final centrifugation was performed under the same conditions, and the supernatants were collected for measurement using the Nanodrop system, quantifying total proteins at 280 nm and DNA at 260 nm.

#### 4.4.3. Measurement of Reactive Oxygen Species

Reactive oxygen species (ROS) production in bacterial cells was detected using the fluorescent molecular probe 2′,7′-dichlorodihydrofluorescein diacetate (H2DCFDA). This probe is cell-permeable, passively diffusing through the cell membrane. Once inside, intracellular esterases cleave the acetate groups, forming the dihydrodichlorofluorescein (H2-DCF) product, which is minimally fluorescent. Upon oxidation by ROS, H2-DCF is converted into 2′,7′-dichlorofluorescein (DCF), a highly fluorescent compound, with fluorescence intensity increasing in correlation with ROS levels.

An overnight culture of *E. coli* was centrifuged at 8000 rpm for 10 min at 22 °C and resuspended in PBS. The bacterial suspension was then adjusted to a turbidity equivalent to 0.5 on the McFarland scale. Subsequently, the suspension was centrifuged again under the same conditions, and the resulting pellet was resuspended with the molecular probe. The probe-labeled bacterial cells were covered with aluminum foil and incubated in the dark at 37 °C for 45 min. After incubation, the cells were centrifuged twice (8000 rpm, 10 min at 22 °C) and resuspended in PBS. Then, 100 μL of the treatments and 100 μL of the probe-labeled bacterial cells were transferred into a black 96-well plate in quadruplicates. Hydrogen peroxide (H_2_O_2_, 0.24%) was used as the positive control, and PBS as the negative control. Fluorescence readings were carried out at 30, 60, 90, and 120 min using a fluorescence spectrophotometer (GloMax microplate reader, Promega, Madison, WI, USA, EUA) with an excitation wavelength of 492–495 nm and an emission wavelength of 517–527 nm. The plate was incubated at 37 °C throughout the fluorescence readings.

#### 4.4.4. Efflux Pump Inhibition

The resistance modulation assay [45,46] evaluated whether the antimicrobials under investigation affected efflux pump activity. *E. coli* XL1-Blue, which is resistant to tetracycline due to the presence of efflux pumps, and *E. coli* K12, which is sensitive to tetracycline, were used in this assay. The MIC of tetracycline was determined both in the presence of combination 1, combination 2, and Bio-AgNP at 1/8 of their MICs, as well as in the absence of these antimicrobials. Efflux pump inhibition was observed when the MIC of tetracycline, in combination with the test compounds, was lower than the MIC of tetracycline alone. MH broth treated with tetracycline was used as the negative control, and bacterial cell viability was monitored throughout the experiment.

### 4.5. AgNP–Antibiotic Interaction Characterization by Optical Analysis

Fourier Transform Infrared (FTIR) analysis of the combinations and individual compounds was performed to identify potential chemical bonds between Bio-AgNP and conventional antibiotics. The analysis was carried out using a Bruker Vortex 70, in a range from 4000 to 400 cm^−1^. The concentrations used were as follows: Bio-AgNP at 10 mM; ampicillin at 20 mg/mL; enrofloxacin at 0.1 mg/mL; combination 1 (5 mM Bio-AgNP + 10 mg/mL ampicillin); and combination 2 (5 mM Bio-AgNP + 0.05 mg/mL enrofloxacin).

### 4.6. Statistical Analysis

Statistical analysis was performed using GraphPad Prism 9 software. First, the data were assessed for normality using the Shapiro–Wilk test and homogeneity of variances using Bartlett’s test. Data that did not meet normality or homogeneity were transformed until they fit the assumptions. Subsequently, analysis of variance (ANOVA) was conducted, followed by Dunnett’s multiple comparison test to identify which groups differed statistically. A significant level of 5% was used, and *p*-values lower than 0.05 were considered statistically significant.

## 5. Conclusions

The growing concern about bacterial resistance has spurred a continuous search for new antimicrobials and innovative strategies. This study contributes to this effort by exploring the combination of conventional antibiotics with nanotechnology, which has emerged as a promising approach to enhance treatment efficacy.

Ampicillin and enrofloxacin, widely used in both human and veterinary medicine, have well-established effectiveness. However, the rise of resistant strains highlights the urgent need for complementary strategies. One such strategy investigated in this study is the combination with Bio-AgNP, whose complementary mechanisms of action enhance the overall therapeutic outcome. The tested combinations showed promising potential in both clinical (Combination 1) and veterinary (Combination 2) applications against MDR isolates.

The prolonged exposure assay to antimicrobial agents, which more closely mimics clinical conditions, revealed that the strains treated with the combinations developed less bacterial tolerance and cross-resistance compared to those treated with individual antimicrobials. When combined, the mechanisms of action of each antimicrobial compound were found to be additive or synergistic, with Bio-AgNP playing a crucial role in disrupting the bacterial membrane and thereby facilitating the entry of antibiotics to their respective intracellular targets. These results corroborate the importance of the rational use of combined therapy to mitigate the development of bacterial resistance.

This study also demonstrated that silver nanoparticles may exhibit antioxidant activity rather than inducing oxidative stress in bacterial cells, hypothesizing that this phenomenon may be attributed to residual antioxidant compounds from the plant extract used in the synthesis.

In conclusion, this study presents promising antimicrobial strategies against multidrug-resistant pathogens, with potential applications in both clinical and veterinary fields. The results highlight the ability of Bio-AgNPs to enhance the efficacy of conventional antibiotics, potentially restoring the activity of drugs previously rendered ineffective by resistance mechanisms. As a preliminary evaluation, this work lays the foundation for future research focused on cytotoxicity assessment, investigation of oxidative stress responses, and in vivo studies to validate the clinical and veterinary applicability of these combinations.

## 6. Patents

This study led to the filing of a patent with the Brazilian Patent and Trademark Office (INPI), under the application number BR 10 2025 015474 9.

## Figures and Tables

**Figure 1 antibiotics-14-00960-f001:**
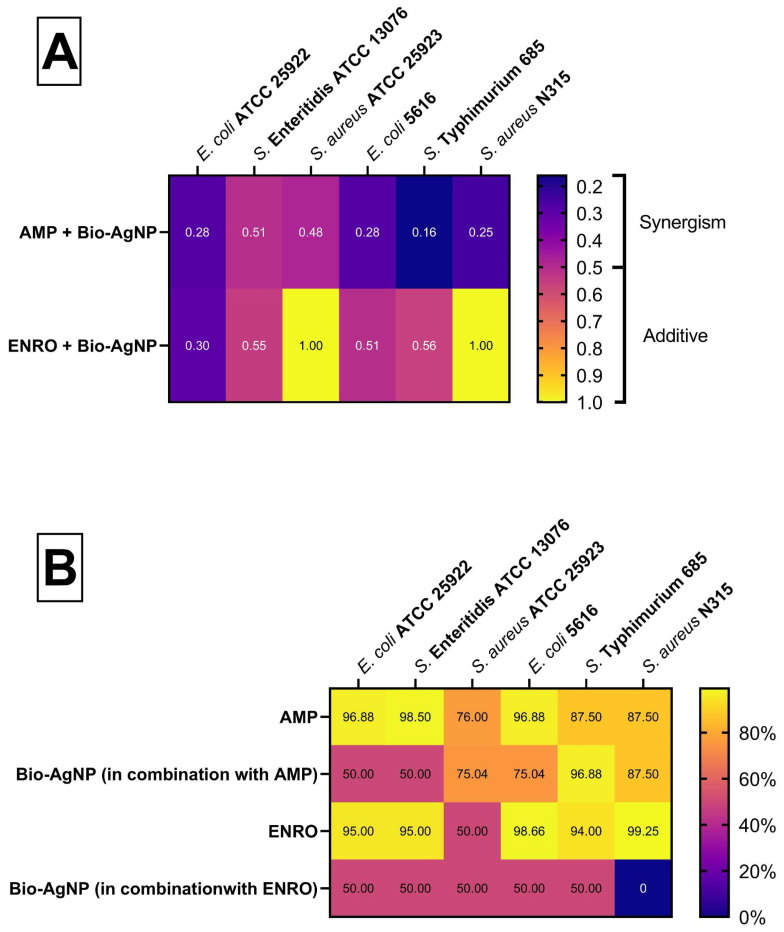
(**A**) Heatmap of the FICI values and antibacterial interaction. (**B**) Heatmap of the percentage reduction in the MIC value of the combinations.

**Figure 2 antibiotics-14-00960-f002:**
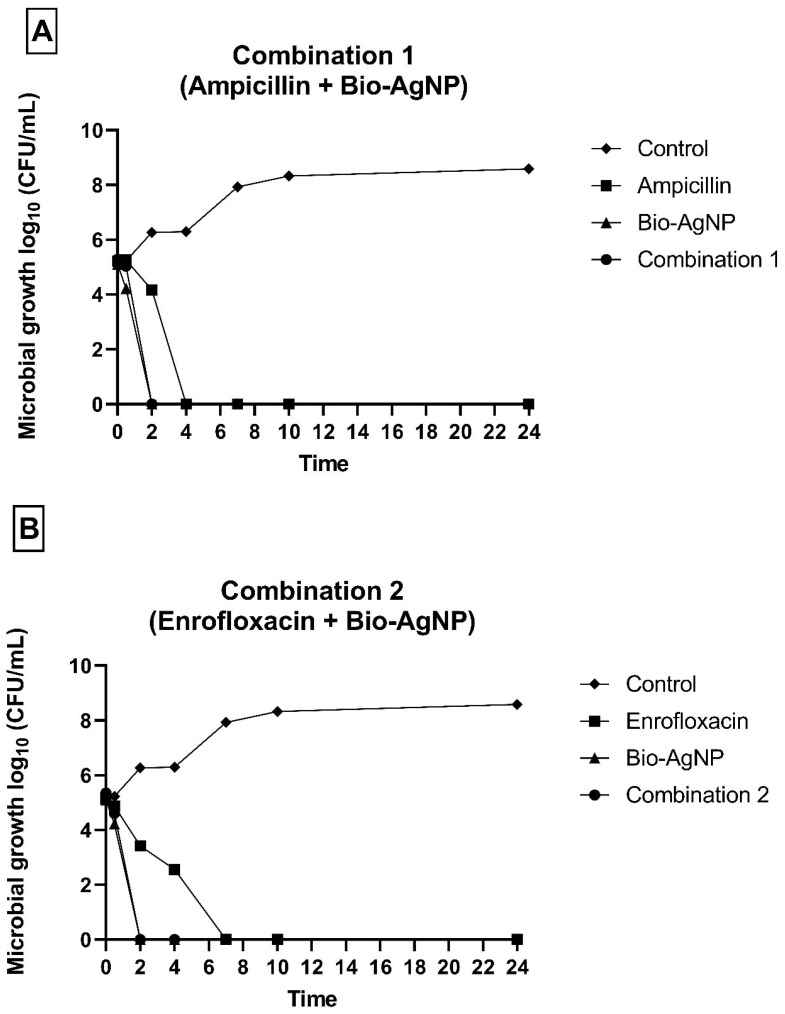
Time–kill assay of *Escherichia coli* treated with (**A**) ampicillin, Bio-AgNP, and combination 1 and with (**B**) enrofloxacin, Bio-AgNP and combination 2.

**Figure 3 antibiotics-14-00960-f003:**
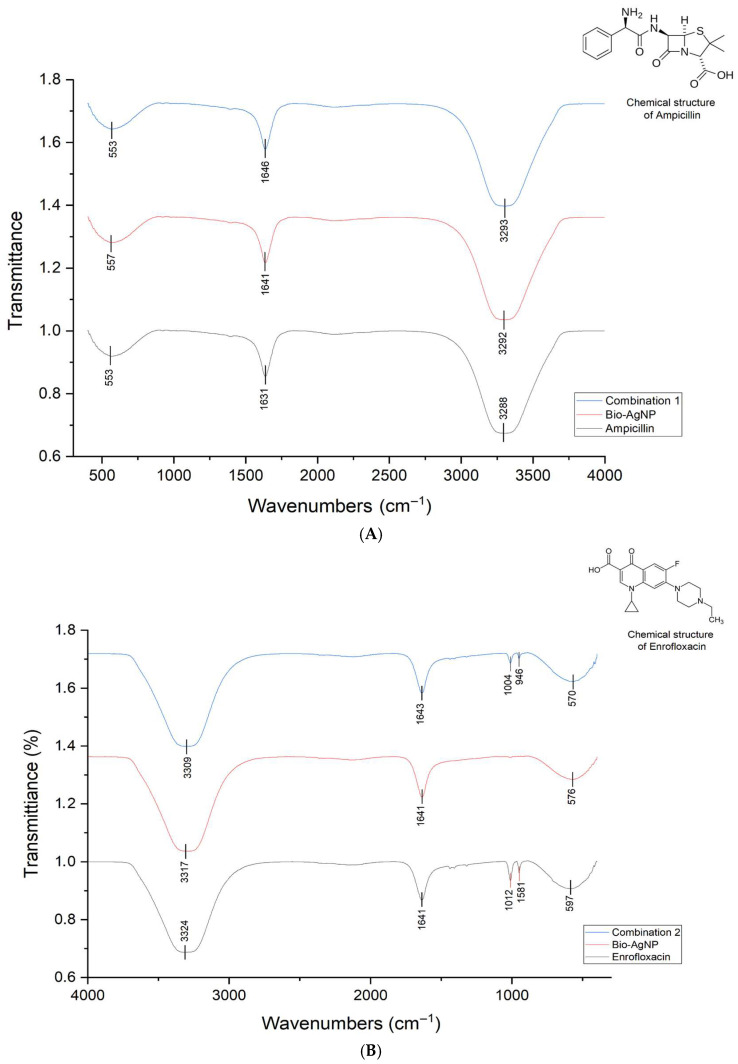
FTIR analysis of Bio-AgNP, antibiotics and combined treatments. (**A**) Spectra of Bio-AgNP, ampicillin and combination 1. (**B**) Spectra of Bio-AgNP, enrofloxacin and combination 2. Transmission spectra of the BioAgNP–antibiotic combination were similar to those of non-combined Bio-AgNP and antibiotics.

**Figure 4 antibiotics-14-00960-f004:**
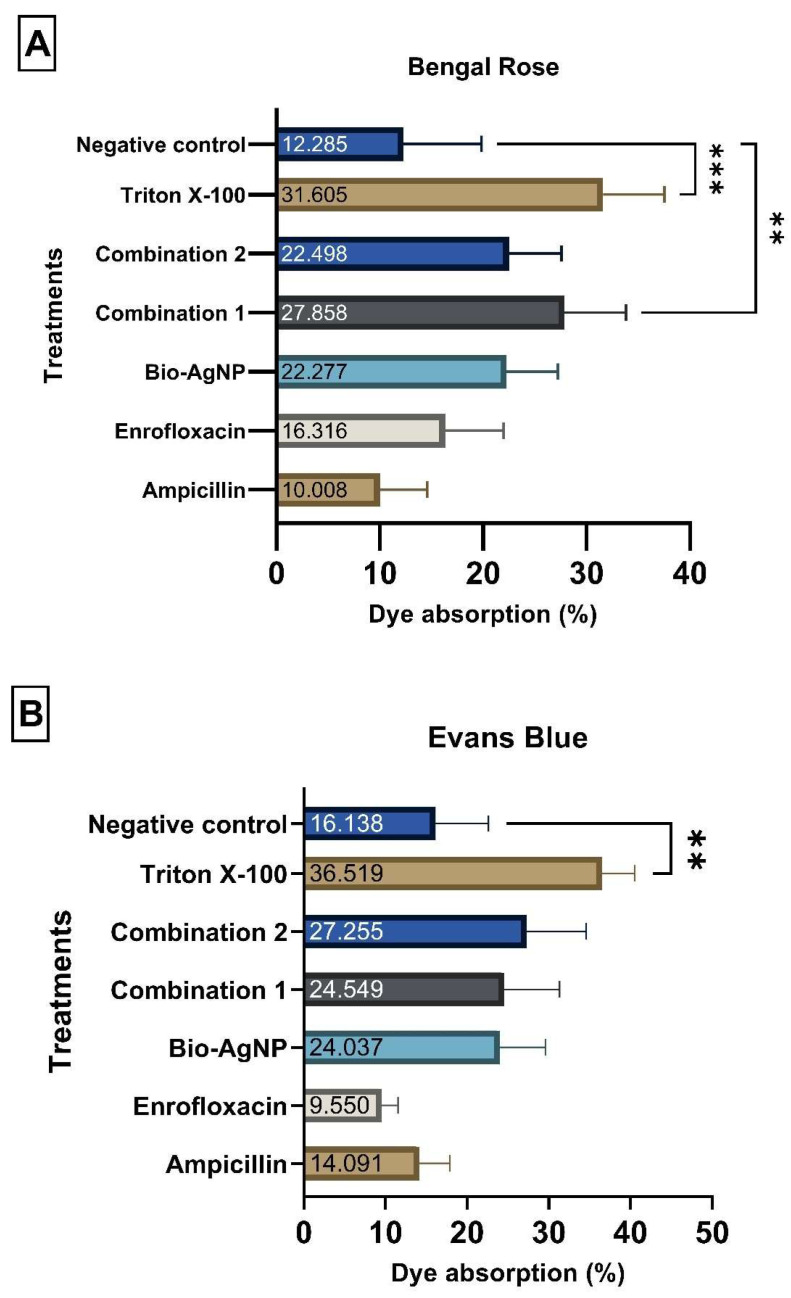
Absorption assay of Rose Bengal (**A**) and Evans Blue (**B**) dyes to analyze changes in membrane permeability in *Escherichia coli* ATCC 25922. **: *p*-value < 0.01. ***: *p*-value < 0.001.

**Figure 5 antibiotics-14-00960-f005:**
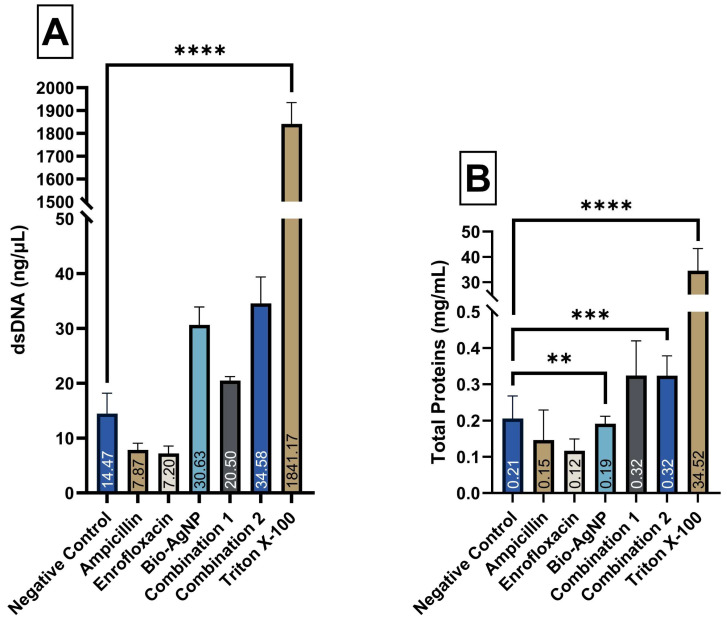
(**A**) Analysis of dsDNA leakage from *Escherichia coli* ATCC 25922 after treatments with Ampicillin, Enrofloxacin, Bio-AgNP, Combination 1, and Combination 2. (**B**) Analysis of Total Protein leakage from *Escherichia coli* ATCC 25922 after treatments with Ampicillin, Enrofloxacin, Bio-AgNP, Combination 1, and Combination 2. The positive control exhibited markedly higher values compared to the other groups. To improve clarity, the y-axis was segmented (discontinuous). Method used for data transformation: (**A**) Square root. (**B**) Negative Logarithmic. **: *p*-value < 0.01. ***: *p*-value < 0.001. ****: *p*-value < 0.0001.

**Figure 6 antibiotics-14-00960-f006:**
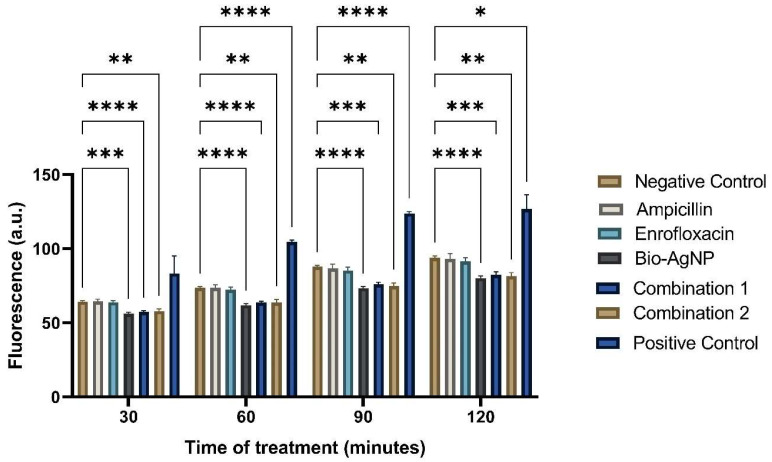
Measurements of oxidative stress were performed by detecting reactive oxygen species (ROS) using molecular probe H_2_DCFDA after exposure to subinhibitory concentrations of antimicrobials at different time points. *: *p*-value < 0.05. **: *p*-value < 0.01. ***: *p*-value < 0.001. ****: *p*-value < 0.0001.

**Table 1 antibiotics-14-00960-t001:** Minimum Inhibitory Concentration and Minimum Bactericidal Concentration of ampicillin (AMP), enrofloxacin (ENRO), and biological silver nanoparticles (Bio-AgNP) against reference and MDR strains.

Bacterial Strain	AMP (μg/mL)	ENRO (μg/mL)	Bio-AgNP (μM)
*E. coli* ATCC 25922	8	0.01	62.5
*S.* Enteritidis ATCC 13076	8	0.03	125
*S. aureus* ATCC 25923	0.5	0.12	62.5
*E. coli* 5616	128	0.015	62.5
*S.* Typhimurium 685	8	0.03	125
*S. aureus* N315	128	0.12	250

**Table 2 antibiotics-14-00960-t002:** Comparison of the MIC value of individual antimicrobial agents before and after prolonged exposure, and the fold increase in MIC.

Bacterial Strain	MIC before Induction of Resistance			MIC after Induction of Resistance			Increase in MIC		
	AMP μg/mL	ENRO μg/mL	Bio-AgNP μM	AMP μg/mL	ENRO μg/mL	Bio-AgNP μM	AMP	ENRO	Bio-AgNP
*E. coli* ATCC 25922	8	0.015	31.2	16	0.32	2000	2×	21.3×	64.1×
*S.* Enteritidis ATCC 13076	8	0.03	125	64	1.92	125	8×	64×	-
*S. aureus* ATCC 25923	0.5	0.12	31.2	20	10	125	40×	83.3×	4×

**Table 3 antibiotics-14-00960-t003:** Comparison of the MIC value of C1 (AMP + Bio-AgNP) before and after prolonged exposure.

Before Induction of Resistance				
Bacterial Strain	AMP (μg/mL)	Bio-AgNP (μM)		
	MIC in Combination	MIC in Combination	FICI	Antibacterial Interaction
*E. coli* ATCC 25922	0.25	15.6	0.53	Additive
*S.* Enteritidis ATCC 13076	0.12	62.5	0.51	Additive
*S. aureus* ATCC 25923	0.12	15.6	0.48	Synergism
**After Induction of Resistance**				
*E. coli* ATCC 25922	1	15.6	0.55	Additive
*S.* Enteritidis ATCC 13076	8	31.2	0.75	Additive
*S. aureus* ATCC 25923	0.12	62.5	0.98	Additive

**Table 4 antibiotics-14-00960-t004:** Comparison of the MIC value of C2 (ENRO + Bio-AgNP) before and after prolonged exposure.

Before Induction of Resistance				
Bacterial Strain	ENRO (μg/mL)	Bio-AgNP (μM)		
	MIC in Combination	MIC in Combination	FICI	Antibacterial Interaction
*E. coli* ATCC 25922	0.0005	15.6	0.55	Additive
*S.* Enteritidis ATCC 13076	0.003	62.5	0.55	Additive
*S. aureus* ATCC 25923	0.06	31.2	1	Additive
**After Induction of Resistance**				
*E. coli* ATCC 25922	0.001	62.5	0.51	Additive
*S.* Enteritidis ATCC 13076	0.06	62.5	0.74	Additive
*S. aureus* ATCC 25923	0.5	31.2	0.98	Additive

**Table 5 antibiotics-14-00960-t005:** Susceptibility profile of *E. coli* ATCC 25922 to antibiotics after prolonged exposure to treatments.

Antibiotic	AMP (mm)	ENRO (mm)	Bio-AgNP (mm)	C1 (mm)	C2 (mm)	S/R (mm)
Amoxicillin-clavulanate	18 S	20 S	26 S	24 S	22 S	≥18/≤13
Ampicillin	10 R	20 S	20 S	16 S	10 R	≥17/≤13
Cefazolin	21 I	25 S	16 R	22 I	11 R	≥23/≤19
Cefepime	31 S	32 S	30 S	30 S	25 S	≥25/≤18
Cefoxitin	25 S	27 S	20 S	25 S	25 S	≥18/≤14
Ceftriaxone	30 S	30 S	30 S	30 S	30 S	≥26/≤22
Ciprofloxacin	40 S	33 S	38 S	40 S	34 S	≥26/≤21
Chloramphenicol	30 S	25 S	30 S	22 S	18 S	≥18/≤12
Enrofloxacin	31 S	20 I	37 S	35 S	26 S	≥23/≤16
Fosfomycin	25 S	27 S	24 S	25 S	25 S	≥16/≤12
Gentamicin	27 S	25 S	30 S	28 S	27 S	≥15/≤12
Imipenem	32 S	30 S	32 S	30 S	34 S	≥23/≤19
Nitrofurantoin	25 S	10 R	25 S	20 S	24 S	≥17/≤14
Sulfamethoxazole/Trimethoprim	27 S	30 S	31 S	27 S	21 S	≥16/≤10
Tetracycline	25 S	25 S	25 S	27 S	23 S	≥15/≤11
Tobramycin	23 S	20 S	20 S	25 S	25 S	≥15/≤12

**Table 6 antibiotics-14-00960-t006:** Susceptibility profile of *S.* Enteritidis ATCC 13076 to antibiotics after prolonged exposure to treatments.

Antibiotic	AMP (mm)	ENRO (mm)	Bio-AgNP (mm)	C1 (mm)	C2 (mm)	S/R (mm)
Amoxicillin-clavulanate	22 S	28 S	25 S	28 S	30 S	≥18/≤13
Ampicillin	10 R	22 S	24 S	10 R	20 S	≥17/≤13
Cefazolin	10 R	24 S	20 I	12 R	25 S	≥23/≤19
Cefepime	24 I	30 S	35 S	25 S	34 S	≥25/≤18
Cefoxitin	10 R	24 S	20 S	10 R	25 S	≥18/≤14
Ceftriaxone	30 S	32 S	30 S	24 I	30 S	≥26/≤22
Ciprofloxacin	30 S	22 I	40 S	32 S	30 S	≥26/≤21
Chloramphenicol	14 I	24 S	30 S	12 R	26 S	≥18/≤12
Enrofloxacin	25 S	16 R	30 S	22 I	30 S	≥23/≤16
Fosfomycin	24 S	30 S	26 S	30 S	30 S	≥16/≤12
Gentamicin	17 S	25 S	25 S	12 R	30 S	≥15/≤12
Imipenem	30 S	35 S	30 S	35 S	33 S	≥23/≤19
Nitrofurantoin	20 S	19 S	19 S	22 S	22 S	≥17/≤14
Sulfamethoxazole/Trimethoprim	20 S	28 S	29 S	25 S	30 S	≥16/≤10
Tetracycline	20 S	20 S	21 S	21 S	23 S	≥15/≤11
Tobramycin	16 I	22 S	20 S	11 R	20 S	≥15/≤12

**Table 7 antibiotics-14-00960-t007:** Susceptibility profile of *S. aureus* ATCC 25923 to antibiotics after prolonged exposure to treatments.

Antibiotic	AMP (mm)	ENRO (mm)	Bio-AgNP (mm)	C1 (mm)	C2 (mm)	S/R (mm)
Ampicillin	0 R	40 S	40 S	20 S	40 S	≥18/≤18
Azithromycin	0 R	25 S	25 S	25 S	25 S	≥18/≤13
Cefoxitin	0 R	30 S	25 S	32 S	30 S	≥22/≤21
Ciprofloxacin	30 S	0 R	27 S	30 S	25 S	≥21/≤15
Clindamycin	0 R	30 S	10 R	20 R	30 S	≥21/≤14
Chloramphenicol	15 I	30 S	30 S	30 S	30 S	≥18/≤12
Enrofloxacin	25 S	10 R	28 S	30 S	25 S	≥23/≤16
Gentamicin	25 S	30 S	30 S	30 S	30 S	≥15/≤12
Nitrofurantoin	0 R	25 S	25 S	30 S	25 S	≥17/≤14
Sulfamethoxazole-Trimethoprim	0 R	30 S	26 S	35 S	35 S	≥16/≤10
Tetracycline	20 R	30 S	30 S	35 S	30 S	≥19/≤14
Vancomycin	0 R	25 S	20 S	20 S	20 S	-

## Data Availability

The original contributions presented in this study are included in the article. Further inquiries can be directed to the corresponding author.

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
