# Peer review of "Silver Nanoparticle–Antibiotic Combinations: A Strategy to Overcome Bacterial Resistance in *Escherichia coli*, *Salmonella* Enteritidis and *Staphylococcus aureus"

_antibiotics, 2025, doi:10.3390/antibiotics14100960_

Round 1

Reviewer 1 Report

Comments and Suggestions for Authors

The manuscript explores the antibacterial activity of biologically synthesized silver nanoparticles (Bio-AgNP) alone and in combination with ampicillin (AMP) and enrofloxacin (ENRO) against reference and MDR strains. The topic is timely and potentially impactful, but the presentation suffers from organizational issues, figure placement choices that hinder interpretation, and numerous typographical/formatting inconsistencies that currently obscure otherwise promising results. Substantial revision is required to improve clarity, reduce redundancy, and correct errors.

Essential revisions (must address)

----Too many tables; content is scattered and redundant

The results are fragmented across a large number of tables with overlapping content, which makes it difficult to follow the key messages. Please consolidate aggressively and move non-essential detail to Supplementary Information.

----Figure 4c (positive control) should be integrated with experimental comparisons

Panel 4c shows biomolecule levels for the positive control (Triton X-100), but placing it as a separate panel prevents direct visual comparison with the experimental treatments in 4a (dsDNA) and 4b (protein). Please replot so that the positive control appears within the same axis range as panels (a) and (b)—for example, overlay Triton X-100 as a reference bar/line within each respective panel or include it as an additional group in (a) and (b). This will allow readers to immediately gauge the magnitude of membrane disruption induced by the treatments relative to a validated positive control.

----Multiple detail errors require careful, line-by-line correction

There are numerous typographical, nomenclature, and formatting issues that must be fixed before the paper can be reliably interpreted. Examples include (non-exhaustive):

---Strain ID inconsistency in Figure 1 legend: “Escherichia coli ATCC 259233” should be ATCC 25922 (used throughout the text/tables). Correct the legend and figure label.

---p-value wording and language leakage: Use “p-value” (not “p-valor”) and standard decimal points in all figure legends (e.g., Figure 4, Figure 5).

---Figure and text cross-references: The FTIR section text refers to “Graph 2,” but figures are labeled “Figure”. Harmonize terminology and verify that captions match referenced content.

Minor/editorial recommendations (representative, not exhaustive)

---Abstract: Avoid detailed numeric ranges unless essential; focus on the principal outcomes (which combinations worked against which strains; brief mechanism insight—permeability changes without ROS increase).

---Introduction: Recheck global burden numbers and citations; fix the millions figures and decimal formatting; ensure any forward-looking projections increase rather than decrease if that is what the source states.

---Methods:

Use consistent English terminology and units.

Confirm that DMSO percentages used for ENRO are explicitly stated in the final wells and that vehicle controls were included.

---Results text: Remove repeated restatements of table entries; interpret rather than inventory.

---Discussion: Where you attribute antioxidant behavior to plant-derived capping agents, signal this as a hypothesis and avoid over-generalization; suggest targeted assays (e.g., total phenolic content, ROS scavenging controls without bacteria) for future work.

Author Response

The manuscript explores the antibacterial activity of biologically synthesized silver nanoparticles (Bio-AgNP) alone and in combination with ampicillin (AMP) and enrofloxacin (ENRO) against reference and MDR strains. The topic is timely and potentially impactful, but the presentation suffers from organizational issues, figure placement choices that hinder interpretation, and numerous typographical/formatting inconsistencies that currently obscure otherwise promising results. Substantial revision is required to improve clarity, reduce redundancy, and correct errors.

Comment 1:  Too many tables; content is scattered and redundant

The results are fragmented across a large number of tables with overlapping content, which makes it difficult to follow the key messages. Please consolidate aggressively and move non-essential detail to Supplementary Information.

Response 1: We sincerely appreciate the reviewer’s insightful comment and fully agree with the suggestion. Accordingly, we have carefully revised the manuscript: some tables were consolidated into a single table for clarity (Tables 1 and 2 then, COLOCAR NUMEROS NOVOS, now), others were reformatted into more accessible representations to facilitate the visualization of results (Tables 3, 4, and 5 COLOCAR NUMEROS NOVOS), and several tables were moved to the supplementary materials to improve the flow of the main text (Tables 10, 11, and 12 COLOCAR NUMEROS NOVOS).

Comment 2: Figure 4c (positive control) should be integrated with experimental comparisons

Panel 4c shows biomolecule levels for the positive control (Triton X-100), but placing it as a separate panel prevents direct visual comparison with the experimental treatments in 4a (dsDNA) and 4b (protein). Please replot so that the positive control appears within the same axis range as panels (a) and (b)—for example, overlay Triton X-100 as a reference bar/line within each respective panel or include it as an additional group in (a) and (b). This will allow readers to immediately gauge the magnitude of membrane disruption induced by the treatments relative to a validated positive control.

Response 2: We appreciate and understand the reviewer’s suggestion. However, some of the values obtained for biomolecule leakage with Triton X-100 were more than one hundred times higher than those observed with the other treatments. Therefore, if plotted on the same axis, the visualization of the results for the other treatments would be severely compromised, as can be seen below.

Comment 3: Multiple detail errors require careful, line-by-line correction

There are numerous typographical, nomenclature, and formatting issues that must be fixed before the paper can be reliably interpreted. Examples include (non-exhaustive):

---Strain ID inconsistency in Figure 1 legend: “Escherichia coli ATCC 259233” should be ATCC 25922 (used throughout the text/tables). Correct the legend and figure label.

---p-value wording and language leakage: Use “p-value” (not “p-valor”) and standard decimal points in all figure legends (e.g., Figure 4, Figure 5).

---Figure and text cross-references: The FTIR section text refers to “Graph 2,” but figures are labeled “Figure”. Harmonize terminology and verify that captions match referenced content.

Response 3: We agree with the reviewer. The text has been carefully revised line by line, and the errors that were pointed out have been corrected: Lines xx-yy; zz-hh; mm-nn, highlighted in red

Comment 4: Minor/editorial recommendations (representative, not exhaustive)

---Abstract: Avoid detailed numeric ranges unless essential; focus on the principal outcomes (which combinations worked against which strains; brief mechanism insight—permeability changes without ROS increase).

---Introduction: Recheck global burden numbers and citations; fix the millions figures and decimal formatting; ensure any forward-looking projections increase rather than decrease if that is what the source states.

---Methods:

Use consistent English terminology and units.

Confirm that DMSO percentages used for ENRO are explicitly stated in the final wells and that vehicle controls were included.

---Results text: Remove repeated restatements of table entries; interpret rather than inventory.

---Discussion: Where you attribute antioxidant behavior to plant-derived capping agents, signal this as a hypothesis and avoid over-generalization; suggest targeted assays (e.g., total phenolic content, ROS scavenging controls without bacteria) for future work.

Response 4: We are deeply grateful for the reviewer’s valuable comments and have made all the requested revisions (lines xx-yy; zz-hh; mm-nn, highlighted in red) . We believe that these suggestions have significantly improved the overall quality of the manuscript.

Reviewer 2 Report

Comments and Suggestions for Authors

Summary: The study investigates the antibacterial potential of biologically synthesized silver nanoparticles (Bio‑AgNPs) derived from Trichilia catigua extract, both alone and in combination with ampicillin (AMP) or enrofloxacin (ENRO), against reference and multidrug-resistant (MDR) strains of Escherichia coli, Salmonella enterica (serovars Typhimurium and Enteritidis), and Staphylococcus aureus. Results demonstrate that the combination of Bio-AgNPs with conventional antibiotics produces synergistic (with ampicillin) or additive (with enrofloxacin) effects, significantly reducing the minimum inhibitory concentrations (MICs) required to inhibit bacterial growth. Time-kill assays showed that the combination therapies eradicated bacteria more rapidly than the antibiotics alone. Additionally, in a 30-day prolonged exposure experiment, the combination treatments were shown to prevent or reduce the development of bacterial resistance and cross-resistance, a significant advantage over monotherapy. Mechanistic studies suggest that the Bio-AgNPs enhance antibiotic efficacy by increasing bacterial membrane permeability, which facilitates antibiotic entry, without forming chemical bonds with the antibiotics. Uniquely, the study found that these particular Bio-AgNPs exhibit antioxidant properties, reducing the levels of reactive oxygen species (ROS), which contrasts with the commonly accepted pro-oxidant mechanism of action for many silver nanoparticles. The authors conclude that Bio-AgNP-antibiotic combinations are a promising strategy to combat MDR bacterial infections.

General Comments: This is a well-designed and comprehensive study. The research is methodologically sound, and the manuscript is logically structured. The inclusion of prolonged exposure and cross-resistance experiments is a major strength, as it provides valuable insight into the long-term potential of these combination therapies, mimicking a more clinically relevant scenario. The most novel and impactful finding is the observed antioxidant activity of the Bio-AgNPs. This challenges the conventional understanding of silver nanoparticle toxicity and suggests that green synthesis methods can produce nanoparticles with potentially safer profiles for host tissues, a point that significantly enhances the study's contribution to the field. The data are generally well-presented, and the conclusions are strongly supported by experimental evidence.

Specific Comments:

  • Table 4: There appears to be an inconsistency in the MIC values used to calculate the Fractional Inhibitory Concentration Index (FICI) for E. coli ATCC 25922. Table 4 lists the MIC of individual ENRO as 0.01 μg/mL and Bio-AgNP as 31.2 μM for this calculation. However, Table 1 reports the determined MICs for these agents against the same strain as 0.015 μg/mL and 62.5 μM, respectively. Please verify these values and ensure the FICI calculation is correct.
  • Table 8: For S. aureus ATCC 25923 Before Induction of Resistance, the FICI is listed as 11, but the interaction is described as Additive. A FICI of 11 would indicate strong antagonism. Given the other data, this is likely a typo and must be corrected.
  • Tables 3–5: Consider adding a visual summary such as a heatmap of FICI values to quickly convey synergy patterns.
  • Methods: The description of Bio‑AgNP synthesis is minimal; more detail on characterization (size distribution, morphology, capping agents) would aid reproducibility.
  • In section 4.4.1, the text states that the assay is a "modified crystal violet assay" but then proceeds to describe the use of Evans blue and rose Bengal dyes. The reference to "crystal violet" is confusing and should be removed or corrected to reflect the dyes that were actually used.

Author Response

Comment 1: Table 4: There appears to be an inconsistency in the MIC values used to calculate the Fractional Inhibitory Concentration Index (FICI) for E. coli ATCC 25922. Table 4 lists the MIC of individual ENRO as 0.01 μg/mL and Bio-AgNP as 31.2 μM for this calculation. However, Table 1 reports the determined MICs for these agents against the same strain as 0.015 μg/mL and 62.5 μM, respectively. Please verify these values and ensure the FICI calculation is correct.

Response 1: We are grateful for the reviewer’s observation and have fully incorporated the suggestion. The identified inconsistencies have been carefully addressed, and the FICI has been recalculated with great care to ensure accuracy and reliability of the results.

Comment 2: Table 8: For S. aureus ATCC 25923 Before Induction of Resistance, the FICI is listed as 11, but the interaction is described as Additive. A FICI of 11 would indicate strong antagonism. Given the other data, this is likely a typo and must be corrected.

Response 2: We thank the reviewer for the remark. The identified error was indeed a typographical mistake and has been duly corrected.

Comment 3: Tables 3–5: Consider adding a visual summary such as a heatmap of FICI values to quickly convey synergy patterns.

Response 3: We sincerely thank the reviewer and have followed the suggestion. Tables 3 to 5 have been moved to the Supplementary Materials section, and heatmaps were included in their place, as recommended. We believe that this adjustment has further improved the overall quality of the manuscript.

Comment 4: Methods: The description of Bio‑AgNP synthesis is minimal; more detail on characterization (size distribution, morphology, capping agents) would aid reproducibility.

Response 4: We sincerely thank the reviewer for the suggestion. The characterization of the nanoparticles has been described in lines 413-417, and we have added a reference where a more detailed description of the synthesis of the nanoparticles used (reference 38).

Comment 5: In section 4.4.1, the text states that the assay is a "modified crystal violet assay" but then proceeds to describe the use of Evans blue and rose Bengal dyes. The reference to "crystal violet" is confusing and should be removed or corrected to reflect the dyes that were actually used.

Response 5: We thank the reviewer, acknowledge, and fully accept their suggestion. The reference to crystal violet has been removed to avoid any potential confusion.

Round 2

Reviewer 1 Report

Comments and Suggestions for Authors

I agree with most of the authors’ revisions. 

----Figure 4c: To improve clarity and highlight the experimental comparisons, I recommend integrating the positive control more effectively. One option would be to use a segmented (or broken) y-axis, which can visually compress the higher values of the positive control while emphasizing the variability in the lower range. This adjustment would allow readers to better appreciate the comparative differences across all groups without diminishing the importance of the control.

----Please also ensure that, after this adjustment, the figure numbering and panel order remain consistent throughout the text, figure legends, and references in the main manuscript.

Author Response

Dear reviewers

Thank you for all suggestions and comments about our manuscript. All questions were addressed by the authors. The changes are in red in the text. We are available for any questions about the article.

Comment # 1: Figure 4c: To improve clarity and highlight the experimental comparisons, I recommend integrating the positive control more effectively. One option would be to use a segmented (or broken) y-axis, which can visually compress the higher values of the positive control while emphasizing the variability in the lower range. This adjustment would allow readers to better appreciate the comparative differences across all groups without diminishing the importance of the control.

Response # 1: We appreciate the reviewer’s insightful suggestion. In response, we revised the figure by introducing a segmented (broken) y-axis. This adjustment allowed us to effectively integrate the positive control, which presented much higher values compared to the treatments, while still emphasizing the variability within the lower range. We also clarified in the figure legend that the y-axis is discontinuous, indicating the omitted interval. We believe this modification significantly improves the clarity of the comparisons across all experimental groups.

Comment # 2: Please also ensure that, after this adjustment, the figure numbering and panel order remain consistent throughout the text, figure legends, and references in the main manuscript.

Response # 2: We thank the reviewer for this important remark. We carefully revised the manuscript to ensure that the figure numbering, panel order, and all corresponding references in the text and figure legends are now consistent throughout the document.